# Genetic Diversity and Virulence Variation of *Metarhizium rileyi* from Infected *Spodoptera frugiperda* in Corn Fields

**DOI:** 10.3390/microorganisms12020264

**Published:** 2024-01-26

**Authors:** Yuejin Peng, Yunhao Yao, Jixin Pang, Teng Di, Guangzu Du, Bin Chen

**Affiliations:** Yunnan State Key Laboratory of Conservation and Utilization of Biological Resources, College of Plant Protection, Yunnan Agricultural University, Kunming 650201, China; 2021053@ynau.edu.cn (Y.P.); yaoyunh@163.com (Y.Y.); pjx19961124@163.com (J.P.); dtn1020@163.com (T.D.); duguangzu1986@163.com (G.D.)

**Keywords:** *Metarhizium rileyi*, genetic diversity, ISSR, molecular identification, biological control, *Spodoptera frugiperda*

## Abstract

*Metarhizium rileyi* is an entomopathogenic fungus that naturally infects the larvae of *Spodoptera frugiperda,* and has biocontrol potential. To explore more natural entomopathogenic fungi resources, a total of 31 strains were isolated from 13 prefectures in Yunnan Province. All the strains were identified using morphology and molecular biology. The genetic diversity of the 31 isolates of *M. rileyi* was analyzed using inter-simple sequence repeat (ISSR) techniques. Seven primers with good polymorphism were selected, and fifty-four distinct amplification sites were obtained by polymerase chain reaction amplification. Among them, 50 were polymorphic sites, and the percentage of polymorphic sites was 94.44%. The thirty-one strains were divided into eight subpopulations according to the regions. The Nei’s gene diversity was 0.2945, and the Shannon information index was 0.4574, indicating that *M. rileyi* had rich genetic diversity. The average total genetic diversity of the subpopulations in the different regions was 0.2962, the gene diversity within the populations was 0.1931, the genetic differentiation coefficient was 0.3482 (>0.25), and the gene flow was 0.9360 (<1). The individual cluster analysis showed that there was no obvious correlation between the genetic diversity of the strains and their geographical origin, which also indicated that the virulence of the strains was not related to their phylogeny. Thus, the genetic distance of the different populations of *M. rileyi* in Yunnan Province was not related to the geographical distance. The virulence of those 32 strains against the 3rd-instar larvae of *S. frugiperda* were varied with the differences in geographical locations. On the 10th day of inoculation, seventeen strains had an insect mortality rate of 70.0%, and seven strains had an insect mortality rate of 100%. The half-lethal times of the *M. rileyi* SZCY201010, XSBN200920, and MDXZ200803 strains against the *S. frugiperda* larvae were less than 4 d. Thus, they have the potential to be developed into fungal insecticidal agents.

## 1. Introduction

The genetic diversity of biological populations is a fundamental aspect of the evaluation and utilization of biological resources. It has been widely used in the study of the diversity of insect and plant pathogens and has also gained significance in the study of the diversity of pathogenic fungi [1]. Notably, entomopathogenic fungi are important pathogens that infect host insects (arthropods), contributing significantly to insect population regulation and the biotransformation of natural biological systems [2]. Currently, more than 7000 strains of entomopathogenic fungi, mainly the *Metarhizium* and *Beauveria* species, have the potential to be developed into fungal insecticides against hundreds of different pests worldwide [3,4], including *Spodoptera litura* (Fabricius, 1775) (Lepidoptera: Noctuidae) [5], *Spodoptera frugiperda* (J.E. Smith, 1797) (Lepidoptera: Noctuidae) [6], mosquitos, and locusts [7,8].

*Metarhizium* is ubiquitous in soils, which is conducive to its biocontrol potential [9]. Commercial formulations of *Metarhizium anisopliae* (Mechnikov) Sorokin are widely available, and development continues to improve their efficacy against a wide range of pests [2]. Consequently, *Metarhizium* is widely used as a biological control agent to control crop pests. *Metarhizium rileyi* (Farlow) Samson, also known as *Nomuraea rileyi* [10], is a widespread filamentous fungus, which plays an indispensable role in the field control of Lepidoptera pests [11]. Unlike most other common fungal insect pathogens that are known to have large epizootic potential, such as *M. anisopliae* and *Beauveria bassiana* (Bals), *M. rileyi* has a narrow host range [12,13]. Recent global outbreaks of fall armyworm (*S. frugiperda*) have resulted in severe damage to corn and other crops [14], with numerous epidemics of *M. rileyi* reported in affected regions [15,16]. Maize, a major food crop in China, has been seriously threatened by *S*. *frugiperda* in China in recent years. As of 30 September 2019, *S. frugiperda* was reported in over 70 nations in the tropical and subtropical regions of the world [17]. However, the accumulation of pesticide residues and the risks associated with pest resistance cannot be avoided [18]. There is, therefore, an urgent need to develop and introduce a biological control agent. There were significant differences in the control effects of different biocontrol strains on *S. frugiperda* [19]. The effect of *M. rileyi* on *S. frugiperda* has been reported to be more prominent in the field [20,21].

Unlike bacterial and viral pathogens that infect insects through the digestive tract, entomopathogenic fungi invade insects through the epidermis [22]. The conidium that is produced by *M. rileyi* adheres to the host surface, germinates, and produces a bud tube that penetrates the host cuticle, and the *M. rileyi* cells are transformed into blastocyst spores or mycelium and colonize the insect’s blood cavity, ultimately resulting in death [23].

Previous research has suggested that *M. rileyi* generally has high genetic variability [24], with significant differences in efficacy against pests among various entomopathogenic fungi isolates that were collected from different geographical locations [25]. The virulence of entomopathogenic fungi is known to be dependent on the fungus’s own growth and development ability and stress resistance. For example, the virulence of *M. rileyi* is not only related to the protective and detoxifying enzymes of *S. frugiperda* but also to its own growth, development, and antioxidant capacity [26]. Therefore, high virulence, rapid growth and development, and strong stress resistance are important indicators for selecting biocontrol fungi resources from nature. Thus, microsatellites, (inter-simple sequence repeat [ISSR] markers) [1], single-strand conformation polymorphisms (SSCPs) [27], and amplified fragment length polymorphism markers (AFLPs) [28] can be used to assess the genetic variability of various isolates and their characteristics [29].

To explore more natural entomopathogenic fungi resources, this study investigated 16 regional maize growing areas in Yunnan Province, China and isolated and purified 31 entomopathogenic fungal resources from maize plants that were affected by *S. frugiperda*. The genetic diversity of *M. rileyi* from the different geographical sources was also analyzed. The relationship between *S. frugiperda* and *M. rileyi* is discussed in our results, which provided more abundant microbial resources for the control of *S. frugiperda* and other pests.

## 2. Materials and Methods

### 2.1. Fungal Strains and Insects

The *M. rileyi* strains that were used in the experiment were collected from maize plants that were affected by *S. frugiperda* in different cities of Yunnan Province, China from July 2020 to September 2021. The distribution of the collection sites is shown in Table 1. The body of the infected insect was cultured in moisture for 5 d until the body was covered with mycelia. A small number of spores were inoculated from the surface of the dead insect onto potato dextrose agar (PDA; potato 200 g, glucose 20 g, agar 15 g, ddH_2_O 1000 mL) solid medium and cultured in a constant temperature incubator at 25 °C, with 70% relative humidity, and under 16L:8D conditions. After the mycelia grew, the single colonies were selected for purification and culture; this was conducted one to two times, and the purified strains with full sporulation were stored in the refrigerator at −80 °C for later use. The *S. frugiperda* that was used for the virulence determination was derived from the 3rd-instar larvae of more than 10 generations that were cultured in the laboratory.

### 2.2. Morphological and Molecular Identification of the Fungi

To conduct the preliminary morphological identification of the isolated strains, the conidia of *M. rileyi* that were growing for 15 d were prepared in a spore suspension of 1.0 × 10^7^ spores/mL and inoculated onto Sabouraud dextrose agar supplemented with yeast extract (SDAY; 1% yeast extract, 4% glucose, 1% peptone, and 1.5% agar) plates. After 4 d, the cake was sampled using a 5 mm hole punch in an area with uniform colony growth, transferred to the center of a new SDAY medium, and cultured in a light incubator at 25 °C, under 16L:8D conditions. After 14 d, the colony growth of the strain was observed, the colony color, shape, texture, and edge shape were recorded, and the colony diameter was measured. The diameter of the spores was observed and measured using an optical microscope. Each independent trial was repeated three times, and there were three replicates per trial.

The purified genomic DNA was extracted using the CTAB method [30]. The ribosomal RNA-internal transcribed spacer (ITS) sequence was amplified by polymerase chain reaction (PCR) using the universal primers ITS1 (5′-TCCGTAGGTGAACCTGCGG-3′) and ITS4 (5′-TCCTCCGCTTATTGATATGC-3′) [31,32]. The PCR products were sent to Shenggong Bioengineering Co. Ltd. (Shanghai, China) for sequencing. The ITS sequencing results were submitted to the NCBI website (https://blast.ncbi.nlm.nih.gov/Blast.cgi, accessed on 5 December 2023) BLAST program, and the strains with multiple strains with corresponding sequence homology were selected using MEGA 7.0 software [33]. The phylogenetic tree was constructed using the maximum likelihood method.

### 2.3. Statistics and Analysis of the Inter-Simple Sequence Repeats and the Population Genetic Distance and Cluster Analysis

Using 13 isolates from different regions as the DNA templates, 15 pairs of ISSR primers (Table 2) were used to amplify the template DNA [34,35]. The reaction mixture consisted of 1 μL of the primers, 12.5 μL of PCR mix, 1–2 μL of the template DNA, and ddH_2_O to make up the total volume. The amplification procedure was as follows: denaturation at 94 °C for 2 min; denaturation at 94 °C for 1 min, annealing at 56 °C for 30 s, and extension at 72 °C for 1 min for 35 cycles; elongation for 6 min at 72 °C. The molecular weight of each sequencing strip was estimated according to the size of the marker, and the amplification site was determined. Then, sequencing gel electrophoresis was conducted using the BandScan 5.0 software (www.seekbio.com/soft/254.html, accessed on 5 July 2023), and the electrophoretic mobility of the bands was compared, with the clear and bright bands being selected. To indicate polymorphism, the bands were recorded as 1 (band) and 0 (no band) to establish a binary data matrix. GenAlEx6.51 was used for data sorting and format conversion. Then, Popgene 32 (www.ualberta.ca/~fyeh/popgene_download.html, accessed on 5 July 2023) was used to calculate the mean number of alleles, effective number of alleles (*N*_e_), Nei’s gene diversity index (*H*_e_), Shannon information diversity index (*I*_s_), and percentage of polymorphic loci. The *N*_e_, *H*_e_, *I*_s_, population genetic diversity within a population, total genetic diversity, genetic differentiation coefficient (*G*_ST_), and gene flow (*Nm*) were obtained from different geographical sources [36,37]. The *G*_ST_ was calculated according to the following equation:*G*_ST_ = (*H*_T_ − *H*_S_)/*H*_T_
where *H*_T_ is the total population heterozygosity, considering all the study populations as a whole, and *H*_S_ is the subpopulation heterozygosity, that is, the average heterozygosity of each population. When 0.05 ≤ *G*_ST_ ≤ 0.15, there is a moderate degree of genetic differentiation among the populations; when 0.15 < *G*_ST_ < 0.25, there is a medium degree of genetic differentiation between the populations; when *G*_ST_ ≥ 0.25, there is a large degree of genetic differentiation between the populations. The *Nm* was calculated using the following equation:*Nm* = 0.25 (1 − *G*_ST_)/*G*_ST_
where *Nm* ≥ 1 indicates smooth gene exchange between the populations; *Nm* < 1 indicates that there is limited gene exchange between the populations. The genetic consistency and the *H*_e_ values among the populations were analyzed using the NTSYSpc 2.10e software. Duncan’s new complex range method was used to compare the differences between the genetic distances. A dendrogram of the clusters among 31 strains of *M. rileyi* based on the unweighted pair group method with arithmetic means (UPGMA) [38].

### 2.4. Virulence Test

A spore suspension of 1.0 × 10^8^ spores/mL was prepared from the conidia of *M. rileyi* that grew for 15 d, and the 3rd-instar larvae were inoculated with an impregnation method for about 10 s. After their removal, the excess water on the surface of the larvae was drained with sterilized filter paper [14]. Then, they were placed in a 12-well cell culture plate with corn leaves and fed in a light incubator at 25 °C, with 70% relative humidity, and under 16L:8D conditions. The leaves of the corn that were used to feed the insects were replaced once a day. The death of the insects was observed at the same time every day, and the number of dead insects was recorded and used to calculate the median lethal time (LT_50_) using Kaplan–Meier analysis. Each independent trial was repeated three times, and the number of insects per trial was 30.

### 2.5. Data Analysis

The observation and measurement results for all the phenotypes of all the insects and fungal strains in the three repeated experiments were analyzed using a one-way analysis of variance (ANOVA) and two-way ANOVA, and the mean values among the fungal strains were compared using a Tukey’s HSD test.

## 3. Results

### 3.1. Morphological Characterization of M. rileyi

The body surfaces of the infected *S. frugiperda* that were collected in natural conditions were covered with white mycelium or green powdery spores. Under the conditions of 25 °C, the culture on the SDAY medium lasted for 14 d, and the single colony diameter of each strain ranged between 17 and 20 mm. The front of the colony was mostly white mycelium, and some of the strains of the colony formed white to green powdery spores. The colonies ranged from flat to slightly convex, they were felt-like or fluviform, and sometimes they had concentric rings. The opposite side of the colony was colorless to yellowish brown or brown, with folds and radiating ridges. The spores were oblong and smooth, ranging in length and width from 2.3 to 3.9 μm and from 1.8 to 3.1 μm, respectively (Table 3).

### 3.2. Molecular Identification of Fungal Strains

Cluster analysis of 31 strains *M. rileyi* was achieved by the maximum likelihood method (Figure 1A). A total of 10 *M. rileyi* strains (FQMD200805, XSBN200920, MDXZ200803, LLZK200810, LPLX200813, SB200807, PPDB201006, SZDT200901, WSWH318103, LCDH200913) were selected as 10 groups with phylogenetic trees representing the visible species. *Metarhizium robertsii* and *Metarhizium flavoviride* were used as the out-group. The above 10 strains were grouped together with different strains of *M. rileyi*. According to the results of the molecular identification, all the collected strains were identified as *M. rileyi* (Figure 1B).

### 3.3. Polymorphism of Inter-Simple Sequence Repeat Amplified Products and the Overall Genetic Diversity of Metarhizium rileyi

Seven pairs of primers with stable amplification results were selected from fifteen ISSR primers (Table 2), including 808, 866, 891, M11, M12, P12, and P25. A total of 54 bands were amplified from 31 strains, of which 51 were polymorphic bands, with a polymorphism percentage of 94.44% (Table 4). Genetic diversity analysis was carried out on the 31 strains of *M. rileyi*, and it was found that the total number of alleles (*N*_a_), *N*_e_, *H*_e_, and *I*_s_ were 2.0000, 1.6079, 0.3579, and 0.5335, respectively, indicating high genetic diversity in this region.

### 3.4. Genetic Differentiation of in the Different Geographical Populations of M. rileyi

The tested strains were divided into 13 groups according to the different geographical sources to analyze their genetic diversity indicators (Table 5). The sample number of five groups, including Dehong and Yuxi, only included one strain, and the representativeness could not be determined. Therefore, the genetic diversity analysis was only performed on the data of the other eight groups. The results showed that the *N*_a_, *N*_e_, *H*_e_, and *I*_s_ were 2.0000, 1.4722, 0.2945, and 0.4574 in Kunming, Qujing, Lincang, Puer, Baoshan, Honghe, Xishuangbanna, and Dali, respectively. The *H*_e_ and *I*_s_ were 0.2413 and 0.3669, respectively, indicating a high genetic diversity. In the Puer habitat, the *H*_e_ and *I*_s_ were smaller, being 0.0926 and 0.1284, respectively, indicating low genetic diversity.

The results of the genetic differentiation of the different geographic populations showed that the mean value of the total genetic diversity (*H*_T_) was 0.2962, and the mean value of the genetic diversity within a population (*H*_S_) was 0.1931 (Table 6). The *G*_ST_ was 0.3482, which was greater than the highly differentiated value of 0.25, and the *Nm* was 0.9360 (<1). The results indicated that the gene exchange among the different populations of *M. rileyi* in Yunnan Province was weak. However, the influence of genetic drift could have led to the increase in genetic differentiation among the populations. In terms of genetic variation, 34.82% was caused by inter-group variation, and 65.18% was caused by intra-group variation. The results indicated that the microecological environment of *M. rileyi* varied greatly in the different geographical populations and habitats.

The multiple comparisons (Table 7) showed that there were differences in genetic similarity among the different geographic populations, ranging from 0.7821 (Banna-Honghe) to 0.9245 (Baoshan–Qujing). The genetic distance between Baoshan and Qujing was the smallest (0.0785), which was followed by Dali and Baoshan (0.0788). The greatest genetic distance was between Banna and Honghe (0.2458), which was followed by that between Puer and Lincang (0.2203). These results indicated that the genetic distance of *M. rileyi* in Yunnan Province was not correlated with the geographical distance.

### 3.5. Population Genetic Distance and Cluster Analysis

The results of the cluster analysis of the 31 strains of *M. rileyi* showed that they were not clustered together strictly according to geographical origin. The strains in the same area were clustered together except for the Xundian strain in the Kunming population, the Midugonglang strain in the Dali population, and the geographical population strain with only one sample. This indicated that there was no significant correlation between the genetic diversity of the strains and their geographical origin (Figure 2). The thirty-one strains were divided into two large clades. The first group (Clade I) was composed of two strains from the Honghe population, accounting for only 6.5% of the total strains. The remaining twenty-nine strains were allocated into the second largest group (Clade II), and Clade II was further divided into two clades (Clade III and IV), of which Clade III contained three strains from the Xishuangbanna Prefecture.

### 3.6. Differences in the Virulence of Metarhizium rileyi

The results of the virulence test showed that the strains that were collected from the different regions had certain infectivity and pathogenicity on the 3rd-instar larvae of *S. frugiperda*, and there were significant differences in the mortality among the different strains (*F* = 9.25, *p* < 0.05). Through further analysis of the virulence data of the different geographical strains, the mortality rate of the 3rd-instar larvae was 30.0–100.0%. Among them, 17 strains had a mortality rate of 70.0% or above. The mortality rate of the seven strains, including SZCY201010, XSBN200920, MDXZ200803, YYDG200805, LLZK200810, FQMD200805, and SZBLT201010, reached 100%. The median lethal time (LT_50_) of the different strains ranged from 3.49 to 7.78 d (Table 8). Among the 31 strains, XSBN200920 had the best pathogenicity against the 3rd-instar larvae of *S. frugiperda*. The mortality rate reached 100%, the LT_50_ value was 3.7 d, and the rate of rigor was 86.7%.

## 4. Discussion

Biocontrol fungi play a key role in pest population control [39]. Although many entomopathogenic fungi have been isolated and identified across various habitats, unfortunately, there are not many formulations that are available for commercial use in agriculture and forestry due to the limited effectiveness of fungal insecticides and the low efficiency of product conversion [40]. Therefore, it is imperative to investigate insect fungi with biocontrol potential for pest control. This is largely facilitated by the advances in genomics and molecular biology [41].

The ITS region has been widely used for the molecular identification of fungal species, and it has better molecular identification resolution than the β-tubulin and EF-1α regions [42]. Based on ITS region sequence analysis, 31 strains of fungi were identified as *M. rileyi*. Our results indicated that *M. rileyi* from Yunnan Province had high genetic diversity. Most microbial community studies have focused on genetic and functional diversity rather than species diversity [43]. The genetic diversity of entomopathogenic fungi is closely related to their hosts and habitat, but the relationship between them remains enigmatic [44]. In eastern China’s Anhui Province, investigations of *M. rileyi* revealed a lack of significant host specificity among the strains that were isolated from various hosts in the same region [18]. Notably, four ISSR markers were sufficient to detect significant genetic diversity in the Argentine isolate, independent of either geographical origin or host [45]. Our results also confirmed that there was no significant correlation between the genetic diversity of *M. rileyi* in the different geographical populations and habitats and their geographical origin. In addition, the high genetic variation of *M. rileyi* in this study may indicate functional diversity among the isolates, which may include isolates that are effective against specific pests.

The percentage of polymorphic loci is an important index for measuring the level of genetic variation within a population. In this study, 54 bands were amplified from 31 strains, of which 51 were polymorphic (94.44%). A higher percentage of polymorphic loci indicates that the population has a strong ability to adapt to the environment. This was supported by the results of an ISSR analysis of *B. bassiana* isolates, which showed that they had high genotypic diversity and were clustered according to habitat [46]. Our cluster analysis results showed (Figure 2) that the thirty-one strains were divided into two large clades, Clade I (Honghe population) and Clade II. Clade II extended to form Clade III and IV. From the analysis of the climatic factors, the Honghe Prefecture and Xishuangbanna Prefecture in Branch II and Branch III were situated in subtropical plateau monsoon and tropical monsoon climates, respectively. These collection areas exhibited higher annual average temperatures when compared with those of the other sites, which may explain their clustering into a single branch. The regional characteristics of the *M. rileyi* species groups in Yunnan province were very evident. Since Yunnan Province belongs to the southern part of the Qinghai–Tibet Plateau, which is characterized by high mountains and valleys, and this barrier may be the cause of the genetic differences among the different geographic populations.

About 213 strains of entomopathogenic fungi have been isolated from soil samples across four autonomous regions in China. *Metarhizium anisopliae* and *B. bassiana* account for the vast majority [47]. *M. rileyi* is a dimorphic fungus with slow growth and development [48], and its advantages in soil environments or some insect hosts may not be obvious [49]. Notably, this study identified 31 strains of *M. rileyi* from different regions of Yunnan Province, China, and they were all derived from the same host, *S. frugiperda*. These results may indicate that *M. rileyi* has stronger biocontrol potential against *S. frugiperda* than other entomopathogenic fungi. However, it is important to note that the spores in the soil serve as the initial source of the inoculated bacteria that cause insect diseases. Collecting a large number of corpse samples may help to elucidate the mechanism driving the prevalence of *M. anisopliae* in insects [50]. For example, *Isaria javanica* (Bally) Samson & Hywel-Jones and *I. fumosorosea* Wize are common fungi that are widely used for aphid and whitefly control and other fungal insecticides, making them readily isolated from these hosts [51,52]. In addition, the prevalence, distribution, and fungal infection of insect fungal pathogens are affected by weather parameters, such as temperature, rainfall, and humidity [53]. In our study, the dipping method was used to determine the virulence of fungi with 1 × 10^7^/mL conidia against insect larvae. It is reported that the fourth-instar larvae of *S. frugiperda* were inoculated by spraying with *M. rileyi* KNU-Ye-1 conidial suspension (1 × 10^7^/mL), and the mortality was 89% after 7 days [52]. The reason for the difference may be that the method of virulence determination and the concentration of the fungal spore suspension are different. In addition, the age of the insects is also an important reason for the different virulence of pathogenic fungi to the same insect. Previous virulence measurements of *S. frugiperda* at the 1–5th-instar ages showed that the higher the age of the insect larvae, the weaker the virulence of the pathogenic fungi [26]. Therefore, the strong prevalence of *M. rileyi* in regions that are affected by *S. frugiperda* is likely one of the main reasons for its high prevalence in the wild. Notably, as previously reported, *M. rileyi* dominated strains collected from corn plantations rather than *M. roberts* or *B. bassiana* [53], possibly because *M. rileyi* itself also has some potential properties superior to other entomopathogenic fungi in the field. Alternatively, *S. frugiperda* may somehow promote the popularity of *M. rileyi*. Further research is needed to confirm these speculations. In this study, we identified 31 strains of *M. rileyi* from *S. frugiperda*, analyzed the genetic and functional diversity of *M. rileyi*, and provided more abundant microbial resources for the control of *S. frugiperda* and other pests. Nevertheless, the identified entomopathogenic fungal strains with good biocontrol potential should be further developed and utilized to control agricultural pests such as *S. frugiperda.*

## 5. Conclusions

Through morphological and molecular biological identification, 31 strains of fungi that were isolated from 16 regions and prefectures in Yunnan Province were identified as *M. rileyi*. The genetic diversity of *M. rileyi* based on the ISSRs was relatively high. The genetic distance was not correlated with the geographical distance. There was no significant correlation between the genetic diversity and geographical origin of the strains. The 31 strains of *M. rileyi* had different insecticidal effects on the insects, and there were considerable differences in the virulence among the different strains. The SZCY201010, XSBN200920, and MDXZ200803 strains had the best insecticidal effect on the *S. frugiperda* larvae, and their half-lethal time was less than 4 d, which provides a basis for the development of fungal insecticidal preparations.

## Figures and Tables

**Figure 1 microorganisms-12-00264-f001:**
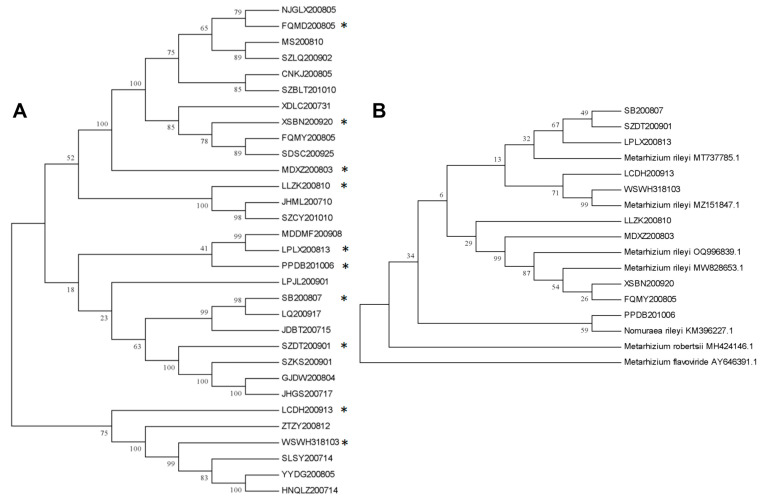
Molecular identification of 31 strains of *M*. *rileyi.* (**A**). Dendrogram built using the unweighted pair group method with the maximum likelihood method which includes 36 isolates of *M. rileyi*. (**B**). Phylogenetic tree of the ITS sequence of 10 *M. rileyi* isolates and their relatives were based on the maximum likelihood method. *M*. *robertsii* and *M*. *flavoviride* were used as the out-group. The asterisks were the strains used to construct the evolutionary tree in (**B**).

**Figure 2 microorganisms-12-00264-f002:**
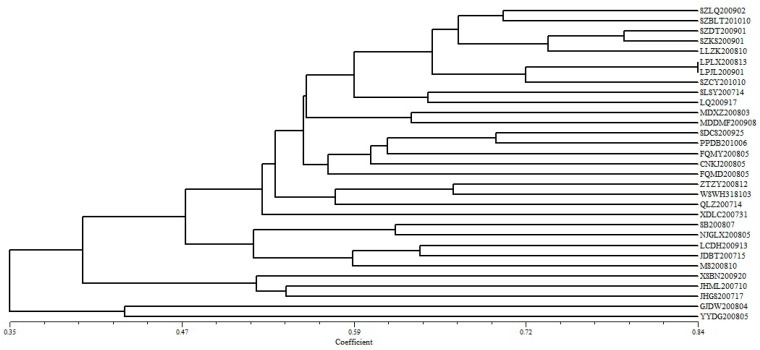
Dendrogram of the clusters among the 31 strains of *M. rileyi* based on the unweighted pair group method with arithmetic means (UPGMA).

**Table 1 microorganisms-12-00264-t001:** Geographic origins of 31 isolates of *Metarhizium* sp.

City	Geographical Origin	Strains	Altitude (m)	Time
Kunming	Xundianlaongchi (103°16′47″ E-25°31′6″ N)	XDLC200731	1843	31 July 2020
	Shilinsuoyishan (103°10′23″ E-24°30′3″ N)	SLSY200714	1688	14 July 2020
	Luquan (102°27′11″ E-24°33′48″ N)	LQ200917	1679	17 September 2020
Qujing	Shizonglongqing (104°46′65″ E-24°34′51″ N)	SZLQ200902	1550	2 September 2020
	Shizongdatong (103°59′38″ E-24°49′31″ N)	SZDT200901	1855	1 September 2020
	Shizongcaiyun (104°5′3″ E-24°29′25″ N)	SZCY201010	1341	10 October 2020
	Shizongboluotang (103°76′44″ E-24°48′58″ N)	SZBLT201010	1976	10 October 2020
	Shizongkuishan (103°73′64″ E-24°46′45″ N)	SZKS200901	1890	3 August 2020
	Luopingluoxiong (104°18′33″ E-24°53′17″ N)	LPLX200813	1454	12 August 2020
	Luopingjiulong (104°43′40″ E-24°63′13″ N)	LPJL200901	1343	1 September 2020
	Luliangzhaokua (24°50′41″ N)	LLZK200810	1994	10 August 2020
Lincang	Fengqingmengyou (99°77′69″ E-24°60′75″ N)	FQMY200805	1396	5 August 2020
	Fengqingmadi (100°4′35″ E-24°29′32″ N)	FQMD200805	2011	5 August 2020
Puer	Lancangdonghui (99°49′38″ E-22°21′47″ N)	LCDH200913	1367	13 September 2020
	Jingdongbeitun (100°22′15″ E-23°56′12″ N)	JDBT200715	2162	15 July 2020
Dali	Nanjiangonglangxaing (100°18′17″ E-24°50′55″ N)	NJGLX200805	1783	4 August 2020
	Miduxizhaung (100°41′19″ E-25°37′23″ N)	MDXZ200803	1726	3 August 2020
	Midudongmafang (100°32′24″ E-25°34′79″ N)	MDDMF200908	1735	8 September 2020
Baoshan	Shidianchangshui (99°5′41″E -25°59′12″ N)	SDSC200925	1920	25 September 2020
	Longyangpiaopu (99°0′51″ E-25°21′17″ N)	PPDB201006	1370	6 October 2020
	Changningkejie (99°26′28″ E-24°51′41″ N)	CNKJ200805	1010	5 August 2020
Xishuangbanna	Xishuangbannan (100°51′29″ E-22°10′22″ N)	XSBN200920	689	20 September 2020
	Jinghongmengla (101°06′12″ E-21°08′3″ N)	JHML200710	1090	10 July 2020
	Jinghonggasa (100°44′53″ E-21°56′42″ N)	JHGS200717	560	17 July 2020
Honghe	Gejiudianwei (103°9′13″ E-23°34′2″ N)	GJDW200804	1130	4 August 2020
	Yuanyangdaguo (102°27′9″ E-22°49′23″ N)	YYDG200805	1450	4 August 2020
Wenshan	Wenshan (104°36′6″ E-23°47′28″ N)	WSWH318103	1967	18 July 2020
Chuxiong	Shuangbai (101°44′34″ E-24°19′54″ N)	SB200807	1183	7 August 2020
Zhaotong	Zhaoyang (103°42′23″ E-27°19′11″ N)	ZTZY200812	1910	12 August 2020
Yuxi	Qinglongzhen (102°53′9″ E-24°19′2″ N)	HNQLZ200714	1124	14 July 2020
Dehong	Mangshi (98°34′39″ E-24°25′41″ N)	MS200810	905	10 August 2020

**Table 2 microorganisms-12-00264-t002:** Inter-simple sequence repeat (ISSR) primers.

Primer	Sequences (5′ to 3′)	Primer	Sequences (5′ to 3′)
807	AGAGAGAGAGAGAGAGT	866	TGCACACACACACAC
808	AGAGAGAGAGAGAGAGC	891	HVHTGTGTGTGTGTGTG
809	AGAGAGAGAGAGAGAGG	M8	CTCCTCCTCCTCCTCCTC
810	GAGAGAGAGAGAGAGAT	M11	AGAGAGAGAGAGAGAGGC
818	CACACACACACACACAG	M12	AGAGAGAGAGAGAGCA
825	ACACACACACACACACT	P12	TGCACACACACACAC
824	GAGAGAGAGAGAGAGAYG	P25	GACAGACAGACAGACAC
834	AGAGAGAGAGAGAGAGYT		

Note: Y = (C/T), H = (A/C/T), V = (A/C/G).

**Table 3 microorganisms-12-00264-t003:** Morphology, colony diameter, and spore size of 31 strains of *M. rileyi* cultured under SDAY plate conditions for 14 days.

Strain No.	Spore Length /μm	Spore Width /μm	Colony Diameter (SDAY)/mm	Colony Morphology
XDLC200731	2.36 ± 0.07	1.82 ± 0.33	17.01 ± 0.41	White, fluffy
SLSY200714	3.36 ± 0.34	2.77 ± 0.44	18.09 ± 0.80	White, fluffy
LQ200917	3.24 ± 0.33	2.13 ± 0.38	17.91 ± 1.37	Light green, powdery, or fluffy
SZLQ200902	3.58 ± 0.19	2.59 ± 0.25	19.02 ± 0.76	Light green, fluffy
SZDT200901	3.52 ± 0.32	2.68 ± 0.44	18.36 ± 1.75	Light green, fluffy
SZCY201010	3.82 ± 0.35	3.11 ± 0.33	19.38 ± 1.22	Light green, powdery, or fluffy
SZBLT201010	3.58 ± 0.32	2.89 ± 0.23	19.37 ± 0.42	Light green, powdery, or fluffy
SZKS200901	3.37 ± 0.41	2.66 ± 0.51	17.24 ± 0.60	White, fluffy
LPLX200813	3.50 ± 0.32	2.80 ± 0.34	18.02 ± 0.77	White, fluffy
LPJL200901	3.17 ± 0.28	2.52 ± 0.48	18.85 ± 0.70	White, fluffy
LLZK200810	3.49 ± 0.23	3.07 ± 0.25	18.34 ± 0.59	White, fluffy
FQMY200805	3.30 ± 0.20	2.64 ± 0.22	18.76 ± 0.61	Light green, powdery, or fluffy
FQMD200805	3.03 ± 0.18	2.54 ± 0.23	18.07 ± 0.88	White, fluffy
LCDH200913	3.28 ± 0.24	2.38 ± 0.48	17.59 ± 0.55	Light green, powdery, or fluffy
JDBT200715	3.44 ± 0.31	2.70 ± 0.46	18.14 ± 0.47	White, fluffy
HNQLZ200714	3.91 ± 0.33	2.85 ± 0.32	18.17 ± 0.89	Light green, powdery, or fluffy
NJGLX200805	3.50 ± 0.23	3.00 ± 0.12	18.24 ± 0.71	White, fluffy
MDXZ200803	3.42 ± 0.56	2.58 ± 0.58	17.32 ± 1.82	White, fluffy
MDDMF200908	3.63 ± 0.17	2.90 ± 0.37	17.57 ± 0.76	Green, powdery or fluffy
SDSC200925	3.51 ± 0.30	2.89 ± 0.47	17.12 ± 0.86	White, fluffy
PPDB201006	3.30 ± 0.46	2.71 ± 0.42	18.37 ± 0.86	White, fluffy
CNKJ200805	3.51 ± 0.18	2.83 ± 0.36	17.87 ± 0.73	White, fluffy
MS200810	3.60 ± 0.36	2.94 ± 0.34	18.10 ± 0.61	Green, powdery, or fluffy
GJDW200804	3.50 ± 0.36	2.60 ± 0.28	17.15 ± 1.23	White, fluffy
YYDG200805	3.55 ± 0.25	2.92 ± 0.29	18.00 ± 0.62	White, fluffy
XSBN200920	3.81 ± 0.24	3.13 ± 0.34	19.78 ± 1.69	Green, powdery
JHML200710	3.58 ± 0.35	2.78 ± 0.41	18.47 ± 0.37	White, fluffy
JHGS200717	3.60 ± 0.22	2.74 ± 0.53	18.86 ± 1.42	Green, powdery, or fluffy
SB200807	3.43 ± 0.20	3.00 ± 0.30	16.83 ± 0.76	Green, powdery, or fluffy
ZTZY200812	3.74 ± 0.15	2.87 ± 0.35	18.70 ± 0.72	Green, powdery, or fluffy
WSWH318103	3.75 ± 0.32	2.80 ± 0.57	17.45 ± 0.60	White, fluffy

**Table 4 microorganisms-12-00264-t004:** Amplification results of the test primers.

Primer Name	Number of Amplified Bands	Number of Polymorphic Bands	Percentage of Polymorphic Loci (%)
808	7	7	100.00%
866	7	6	85.71%
891	10	10	100.00%
M11	8	7	88.89%
M12	8	8	100.00%
P12	5	5	100.00%
P25	9	8	87.50%
Average	54	51	94.44%

**Table 5 microorganisms-12-00264-t005:** Diversity of *M. rileyi* in different geographical populations.

Groups	Simple Size	Na	Ne	He	Is
Kunming	3	0.1500	1.400	0.2222	0.3183
Qujing	8	1.7037	1.3914	0.2413	0.3669
Lincang	2	1.3519	1.2889	0.1759	0.2439
Puer	2	1.1851	1.2778	0.0926	0.1284
Baoshan	3	1.4259	1.3407	0.1893	0.2711
Honghe	2	1.2963	1.2962	0.1481	0.2054
Xishuangbanna	3	1.4074	1.3259	0.1881	0.2539
Dali	3	1.5741	1.4593	0.2551	0.3654
Dehong	1	1.0000	1.0000	0.0000	0.0000
Yuxi	1	1.0000	1.0000	0.0000	0.0000
Chuxiong	1	1.0000	1.0000	0.0000	0.0000
Zhaotong	1	1.0000	1.0000	0.0000	0.0000
Wenshan	1	1.0000	1.0000	0.0000	0.0000
Toal	31	2.0000	1.4722	0.2945	0.4574

**Table 6 microorganisms-12-00264-t006:** Analysis of genetic differentiation among the different geographical populations of *M. rileyi*.

Locus	*H* _T_	*H* _S_	*G* _st_	*Nm*
Mean	0.2962	0.1931	0.3482	0.9360
SD	0.0189	0.0073		

**Table 7 microorganisms-12-00264-t007:** Gene flow and genetic differentiation coefficient between different area groups of *M. rileyi*.

Site	Kunming	Qujing	Lincang	Puer	Baoshan	Honghe	Xishuangbanna	Dali
Kunming		0.8993	0.8074	0.8198	0.8882	0.8359	0.8456	0.8621
Qujing	0.1062		0.8949	0.8883	0.9245	0.8998	0.8762	0.9179
Lincang	0.2139	0.1111		0.8023	0.8351	0.8104	0.8205	0.8135
Puer	0.1987	0.1184	0.2203		0.8535	0.8210	0.8415	0.8354
Baoshan	0.1186	0.0785	0.1802	0.1584		0.8262	0.8686	0.9242
Honghe	0.1792	0.1056	0.2103	0.1972	0.1909		0.7821	0.9004
Xishuangbanna	0.1677	0.1321	0.1979	0.1725	0.1409	0.2458		0.8443
Dali	0.1484	0.0856	0.2065	0.1798	0.0788	0.1050	0.1692	

**Table 8 microorganisms-12-00264-t008:** Pathogenicities of different *M. rileyi* strains against the 3rd-instar larvae of *S. frugiperda* at 1.0 × 10^8^ spore/mL (8 d).

Strain No.	Regression Equation	Correlation Coefficient	Mortality	Rigid Cadaver	LT50 (d)
JHML200710	Y = 0.83 + 5.04X	0.94	76.7 ± 3.3 ^cde^	61.1 ± 6.7 ^abcdef^	6.22
ZTZY200812	Y = −0.45 + 6.23X	0.96	73.3 ± 8.8 ^def^	63.3 ± 12.0 ^abcde^	5.71
WSWH318103	Y = 3.35 + 1.63X	0.89	60.0 ± 5.7 ^efg^	33.3 ± 6.7 ^fgh^	7.49
MDDMF200908	Y = 2.74 + 2.46X	0.92	63.3 ± 6.7 ^efg^	50.0 ± 5.8 ^cdefg^	6.74
GJDW200804	Y = −0.36 + 5.87X	0.96	73.3 ± 12.0 ^def^	43.3 ± 6.7 ^defg^	6.38
SLSY200714	Y = 2.64 + 2.48X	0.95	66.7 ± 3.3 ^defg^	30.0 ± 15.2 ^gh^	7.78
YYDG200805	Y = 2.59 + 5.24X	0.78	100.0 ± 0.0 ^a^	70.0 ± 10.0 ^abcd^	4.27
SZDT200901	Y = 1.13 + 4.63X	0.93	80.0 ± 5.8 ^bcde^	30.0 ± 5.7 ^gh^	5.79
XDLC200731	Y = −1.61 + 5.64X	0.84	30.0 ± 5.7 ^h^	10.0 ± 5.7 ^h^	-
LLZK200810	Y = −0.99 + 8.84X	0.93	100.0 ± 0.0 ^a^	60.0 ± 5.7 ^abcdef^	4.55
JHGS200717	Y = 2.34 + 4.58X	0.92	96.6 ± 3.3 ^ab^	73.3 ± 3.3 ^abc^	4.65
MDXZ200803	Y = −1.58 + 11.89X	0.95	100.0 ± 0.0 ^a^	53.3 ± 14.5 ^bcdefg^	3.49
FQMD200805	Y = 0.38 + 9.09X	0.89	100.0 ± 0.0 ^a^	43.3 ± 12.0 ^defg^	3.73
SZBLT201010	Y = −0.27 + 9.55X	0.95	100.0 ± 0.0 ^a^	86.7 ± 8.8 ^a^	3.78
SZCY201010	Y = 1.65 + 6.59X	0.87	100.0 ± 0.0 ^a^	80.0 ± 15.2 ^ab^	3.95
MS200810	Y = 3.24 + 2.02X	0.99	60.0 ± 5.7 ^efg^	33.3 ± 6.6 ^fgh^	5.96
SDSC200925	Y = 3.23 + 2.29X	0.99	70.0 ± 5.7 ^defg^	36.7 ± 8.8 ^efgh^	5.24
SZLQ200902	Y = 3.66 + 1.67X	0.97	63.3 ± 3.3 ^efg^	30.0 ± 5.7 ^gh^	6.08
XSBN200920	Y = −0.28 + 10.22X	0.94	100.0 ± 0.0 ^a^	86.7 ± 6.7 ^a^	3.69
LQ200917	Y = 3.75 + 1.43X	0.96	53.3 ± 8.8 ^fg^	26.7 ± 6.7 ^gh^	6.85
SZKS200901	Y = 3.46 + 2.08X	0.98	66.7 ± 6.7 ^defg^	30.0 ± 5.7 ^gh^	5.13
LPLX200813	Y = 3.39 + 1.64X	0.92	50.0 ± 5.7 ^g^	30.0 ± 5.7 ^gh^	7.47
SB200807	Y = 3.58 + 1.47X	0.97	53.3 ± 6.6 ^fg^	40.0 ± 5.7 ^efg^	6.78
FQMY200805	Y = 3.72 + 1.79X	0.96	70.0 ± 5.7 ^defg^	36.6 ± 6.7 ^efgh^	5.56
HNQLZ200714	Y = 3.19 + 2.37X	0.99	63.3 ± 8.8 ^fgh^	33.3 ± 7.1 ^efg^	5.57
JDBT200715	Y = 3.62 + 2.29X	0.97	80.0 ± 5.7 ^bcde^	43.3 ± 6.7 ^defg^	4.90
NJGLX200805	Y = 3.22 + 2.42X	0.99	70.0 ± 3.3 ^defg^	36.6 ± 3.3 ^efgh^	5.21
LCDH200913	Y = 1.03 + 4.75X	0.94	66.7 ± 6.7 ^defg^	43.3 ± 8.8 ^defg^	5.94
PPDB201006	Y = 3.09 + 2.66X	0.99	86.7 ± 3.3 ^abcd^	53.3 ± 3.3 ^bcdefg^	5.24
LPJL200901	Y = 2.78 + 2.49X	0.97	50.0 ± 5.7 ^g^	30.0 ± 5.7 ^gh^	7.54
CNKJ200805	Y = 1.32 + 4.53X	0.92	70.0 ± 10.0 ^defg^	36.6 ± 8.8 ^efgh^	5.57

Note: Y means probit, X is log of time. “-” represents that the mortality of the infected 3rd-instar larvae was less than 50.00%, LT_50_ values could not be estimated. The lowercase letters a–h represent the level of significant difference between the groups.

## Data Availability

Data are contained within the article.

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
