# Peer review of "Genetic Diversity and Virulence Variation of Metarhizium rileyi from Infected Spodoptera frugiperda in Corn Fields"

_microorganisms, 2024, doi:10.3390/microorganisms12020264_

Round 1

Reviewer 1 Report

Comments and Suggestions for Authors

The paper appears well planned, relatively well written, and represents a contribution to science literature. This information warrants publication; however, this paper requires moderate revision.

I believe that the authors should report information on the morphological characteristics of the Metarhizium rileyi.

Minor modifications required:

Please make sure the species names are written in italics, also "frugiperda" is common name.

Lines 183-192: With this description a morphological identification cannot be considered. It is necessary to complement it.

Table 1 and 3 They are equal

Table 8 How was LT50 estimated? It does not have independent information to calculate it

I invite the authors to improve the discussion on virulence, considering the stage of the insect and the way of inoculating them, in addition to expanding the part of other isolates from other countries.

Author Response

Thank you very much for your professional advice. We have revised the manuscript according to your requirements and annotations. We hope to obtain your approval and consent. Thank you again

Please make sure the species names are written in italics, also "frugiperda" is common name.

Response: We have checked the full text and completed the revisions in the manuscript.

Lines 183-192: With this description a morphological identification cannot be considered. It is necessary to complement it.

Response: Thank you for your suggestion. When we checked the manuscript, we found duplications in Table 3 and Table 1, and replaced Table 3 with the latest version. The content of this part ( Lines 186-192) is a general description of the morphology of 31 strains of fungi. We think this part of the content needs such a description, and hope to get your consent.

Table 1 and 3 They are equal

Response:We have replaced the correct table 3 in the manuscript

Table 8 How was LT50 estimated? It does not have independent information to calculate it.

Response: We have supplemented the manuscript with information on the calculation of LT50.

I invite the authors to improve the discussion on virulence, considering the stage of the insect and the way of inoculating them, in addition to expanding the part of other isolates from other countries.

Response: We have supplemented in the manuscript (Line 371-380).

Reviewer 2 Report

Comments and Suggestions for Authors

Here are my comments on this ms:

Revise the title by removing "of"

L43 follow the guidelines of the journals in how to cite references, check this issue in whole ms.

L87 DO not localize your study, revise: This is the first report on the genetic diversity of M. rileyi in southwest China.

Add the units in the Tables where are possible.

Disscussion is very week and superficial, the authors should improve it and add the different mechanisms, more references and more details to make it stronger than current version.

Good luck

Author Response

Thank you very much for your professional advice. We have revised the manuscript according to your requirements and annotations. We hope to obtain your approval and consent. Thank you again

Revise the title by removing "of"

Response: We have revised it in the manuscript.

L43 follow the guidelines of the journals in how to cite references, check this issue in whole ms.

Response:We have revised it in the manuscript.

L87 DO not localize your study, revise: This is the first report on the genetic diversity of M. rileyi in southwest China.

Response: We have deleted it in the manuscript

Add the units in the Tables where are possible.

Response: We have revised it in the manuscript

Disscussion is very week and superficial, the authors should improve it and add the different mechanisms, more references and more details to make it stronger than current version.

Response: We have supplemented in the manuscript (Line 323-326, Line 371-380).

Round 2

Reviewer 1 Report

Comments and Suggestions for Authors

The manuscript was corrected as requested